# Steppe lemmings and Chinese hamsters as new potential animal models for the study of the *Leishmania* subgenus *Mundinia* (Kinetoplastida: Trypanosomatidae)

**Tomas Becvar** [iD], **Barbora Vojtkova, Lenka Pacakova, Barbora Vomackova Kykalova, Lucie Ticha, Petr Volf, Jovana Sadlova** [iD] *

Department of Parasitology, Faculty of Science, Charles University, Prague, Czech Republic

* jovanas@seznam.cz

**Data Availability Statement:** The authors confirm that all data underlying the findings are fully

## Abstract

*Leishmania*, the dixenous trypanosomatid parasites, are the causative agents of leishmaniasis currently divided into four subgenera: *Leishmania*, *Viannia*, *Sauroleishmania*, and the recently described *Mundinia*, consisting of six species distributed sporadically all over the world infecting humans and/or animals. These parasites infect various mammalian species and also cause serious human diseases, but their reservoirs are unknown. Thus, adequate laboratory models are needed to enable proper research of *Mundinia* parasites. In this complex study, we compared experimental infections of five *Mundinia* species (*L. enriettii, L. macropodum, L. chancei, L. orientalis*, and four strains of *L. martiniquensis*) in three rodent species: BALB/c mouse, Chinese hamster (*Cricetulus griseus*) and steppe lemming (*Lagurus lagurus*). Culture-derived parasites were inoculated intradermally into the ear pinnae and progress of infection was monitored for 20 weeks, when the tissues and organs of animals were screened for the presence and quantity of *Leishmania*. Xenodiagnoses with *Phlebotomus duboscqi* were performed at weeks 5, 10, 15 and 20 post-infection to test the infectiousness of the animals throughout the experiment. BALB/c mice showed no signs of infection and were not infectious to sand flies, while Chinese hamsters and steppe lemmings proved susceptible to all five species of *Mundinia* tested, showing a wide spectrum of disease signs ranging from asymptomatic to visceral. *Mundinia* induced significantly higher infection rates in steppe lemmings compared to Chinese hamsters, and consequently steppe lemmings were more infectious to sand flies: In all groups tested, they were infectious from the 5th to the 20th week post infection. In conclusion, we identified two rodent species, Chinese hamster (*Cricetulus griseus*) and steppe lemming (*Lagurus lagurus*), as candidates for laboratory models for *Mundinia* allowing detailed studies of these enigmatic parasites. Furthermore, the long-term survival of all *Mundinia* species in steppe lemmings and their infectiousness to vectors support the hypothesis that some rodents have the potential to serve as reservoir hosts for *Mundinia*.

available without restriction. All relevant data are within the paper and its Supporting Information files.

**Funding:** This work was supported by the project "Grant Schemes at CU" at the Univerzita Karlova v Praze (reg. no. 403 CZ.02.2.69/0.0/0.0/19_073/0016935 to TB, BV, BVK, LP and LT) and by the ERD Funds, project CePaV- 404 iP at the Ministerstvo Školství, Mládeže a Tělovýchovy (grant No. CZ.02.1.01/0.0/0.0/16_019/0000759 to JS and PV). This work was supported by European Commission, Horizon 2020 Infrastructure Infravec2 project (https://infravec2.eu/) under grant agreement No 731060. The funders had no role in study design, data collection and analysis, decision to publish, or preparation of the manuscript.

**Competing interests:** The authors have declared that no competing interests exist.

## Author summary

*Leishmania* parasites of the subgenus *Mundinia* are an emerging health and veterinary problem that should not be ignored. Being the most recent of all *Leishmania* described, many aspects of *Mundinia* biology are enigmatic. We have very scarce data on their life cycles and biology, thus proper laboratory research must be done to enable their better understanding. One of the most crucial parts of the life cycle of *Leishmania* is the development in the mammalian host. In the past, we worked on establishment of other laboratory models for the subgenus, but neither *Arvicanthis*, *Mastomys*, nor guinea pigs proved to be a good choice. Other authors performed experiments with BALB/c mice using various inoculation techniques, but they also failed. Here we describe the establishment of two new potential laboratory model species, Chinese hamsters and steppe lemmings, which proved to be susceptible to *Mundinia* and such findings will enable other scientists to continue in research of these parasites.

## 1. Introduction

Leishmaniases are a group of diseases caused by digenetic parasites of the genus *Leishmania* that naturally occur in tropical and subtropical regions around the world. These parasites are currently divided into four subgenera: *Leishmania*, *Viannia*, *Sauroleishmania*, and *Mundinia* [1]. With more than one million human cases a year, leishmaniases are among the world's most important tropical diseases caused by protozoa. There are three main clinical manifestations of the disease: cutaneous, mucocutaneous, and visceral, with symptoms depending on many endogenous and exogenous factors, such as parasite species and virulence, infectious dose, the host immune response, and other biological, socio-economic and environmental factors [2–4].

*Mundinia* is the most recently described *Leishmania* subgenus consisting of six species, all of which have unknown reservoir hosts [5,6]. Biting midges (Diptera: Ceratopogonidae) are the most probable vectors of *Mundinia*, changing the established dogma that *Leishmania* are transmitted solely by phlebotomine sand flies (Diptera: Psychodidae) [7–10]. Three species of *Mundinia* do not infect humans; *L. enriettii* was found in domestic guinea pigs in Brazil, and even though nearly 80 years have passed since its discovery, it remains one of the most enigmatic species of *Leishmania* [11]. *Leishmania macropodum*, which causes cutaneous and mucocutaneous symptoms in kangaroos, is the only representative species of the *Leishmania* genus in Australia [12,13] and *L. procaviensis* has recently been described from hyrax (*Procavia capensis*) isolates in Namibia [6,14].

Other three species of *Mundinia* infect humans; the symptoms of the disease range from a single cutaneous lesion to general visceral infection [15–17]. *Leishmania martiniquensis* is the most widespread species of all known *Leishmania*, present in Central and North America, Europe and Asia [17–20] while *L. orientalis* is endemic to Southeast Asia [16,21–23]. In Thailand, a high prevalence of *L. martiniquensis* and several cases of *L. orientalis* have been described in immunosuppressed people [24], but cases of immunocompetent people showing signs of cutaneous or visceral leishmaniasis have also been reported [25]. The last member of the subgenus is recently described *L. chancei* [6] which causes cutaneous leishmaniasis in humans, mainly children and young adults in Ghana [15]. Based on the low number of human cases, the zoonotic circulation of this parasite is predicted [26].

Several mammalian species are currently incriminated for their potential involvement in *Mundinia* circulation. Black rats (*Rattus rattus*), cows, and horses are suggested candidates,

since *L. martiniquensis* DNA was detected and viable parasites were cultured from these hosts [18–20,27]. However, it should be noted that detection of parasites or their DNA does not imply the reservoir potential of the host, since parasites can be present in the organism incidentally, be detectable, but not transmittable to other hosts. These organisms are so called "sinks" of infection. The opposite are so called "sources", where parasites are detectable and further transmittable to other hosts [28].

In this study, we decided to test susceptibility to *Mundinia* and infectiousness to sand flies by xenodiagnoses in both a classic model laboratory animal, the BALB/c mouse, and two less established model rodent species, steppe lemming (*Lagurus lagurus*) and Chinese hamster (*Cricetulus griseus*). Laboratory mice (*Mus musculus*) have become one of the most widely used biological models in many areas of biomedical research, including the study of cutaneous and visceral leishmaniasis, mainly due to the wide range of inbred strains and transgenic animals. In contrast, *Lagurus lagurus* and *Cricetulus griseus* are genetically polymorphic wild rodents. They were chosen because they are commercially available and easy to maintain and several old publications [29–32] as well as more recent ones [33,34], have demonstrated their high susceptibility to various *Leishmania* parasites.

## 2. Materials and methods

### 2.1. Ethics statement

The animals were maintained and handled in the Charles University animal facility in Prague following institutional guidelines and Czech legislation (Act No. 246/1992 and 359/2012 coll. on Protection of Animals against Cruelty in present statutes at large), which complies with all relevant European Union and international guidelines for experimental animals. All experiments were approved by the Committee on the Ethics of Laboratory Experiments of Charles University in Prague and were performed with permission no. MSMT-7831/2020-3 of the Ministry of Education, Youth, and Sports. The investigators are certified for experimentation with animals by the Ministry of Agriculture of the Czech Republic.

### 2.2. Parasites and rodents

*Leishmania enriettii* (MCAV/BR/45/LV90), *L. macropodum* (MMAC/AU/2004/AM-2004), *L. chancei* (MHOM/GH/2012/GH5), *L. orientalis* (MHOM/TH/2014/LSCM4) and four strains of *L. martiniquensis* (MHOM/MQ/1992/MAR1; MHOM/TH/2011/Cu1R1; MHOM/TH/2019/Cu2 and MEQU/CZ/2019/Aig1) were used. Parasites were maintained at 28°C in M199 medium supplemented with 20% fetal calf serum (Gibco, Prague, Czech Republic), 1% BME vitamins (Sigma-Aldrich, Prague, Czech Republic), 2% sterile urine and 250 μg/ml amikacin (Amikin, Bristol-Myers Squibb, Prague, Czech Republic). All strains were kept in a cryobank with 2–3 subpassages *in vitro* prior to experimental infections of rodents. Before experimental infection, the parasites were washed by centrifugation (1400×g/5 min) and resuspended in saline solution.

BALB/c mice (Velaz, Prague, Czech Republic), steppe lemmings and Chinese hamsters (Karel Kapral s.r.o., Prague, Czech Republic) were maintained in groups of 5 specimens in T4 boxes (58 × 37 × 20 cm) (Velaz, Prague, Czech Republic), equipped with bedding (SubliCZ, Sojovice, Czech Republic), breeding material (Woodwool, Miroslav Vlk s.r.o., Czech Republic) and hay (Krmne smesi Kvidera, Spalene Porici, Czech Republic), provided with a feed mixture ST-1 (Krmne smesi Kvidera, Spalene Porici, Czech Republic) and water *ad libitum*, with a 12 h light/12 h dark photoperiod, temperature of 22–25°C and relative humidity of 50–60%.

## 2.3. Experimental infections and xenodiagnoses of rodents

Eighty specimens of three rodent species (*Mus musculus*, *Cricetulus griseus* and *Lagurus lagurus*) anesthetised with ketamin/xylazin (62 mg/kg and 25 mg/kg) were injected with $10^7$ stationary-stage promastigotes (from 5–7 days old cultures, depending on the species) in 5 μl of sterile saline solution intradermally into the left ear pinnae. The ear pinna was chosen because it is the most appropriate site for monitoring the development of lesions and it is also the site where sand flies naturally bite, therefore suitable for xenodiagnoses [33,35,36]. The course of infection was recorded weekly.

Xenodiagnoses were performed at weeks 5, 10, 15 and 20 p.i. using *Phlebotomus duboscqi*. Five to six-day-old sand flies were placed in plastic vials covered with fine nylon mesh and allowed to feed on the ear pinnae of anesthetised animals. The proven vectors of *Mundinia* are unknown and a laboratory colony of the permissive species *Culicoides sonorensis* (Pirbright institute, UK) is not available for routine experiments in the Czech Republic, but early infections in vectors are nonspecific [37,38], so *Mundinia* can multiply before defecation of the sand fly, which in *P. duboscqi* is on day 4–5 PBM [39]. The engorged individuals were maintained for two days at 25°C and then stored in tissue lysis buffer (Roche, Prague, Czech Republic) at -20°C in pools of 5 females for subsequent DNA isolation and nested PCR. Specimen for xenodiagnoses were chosen randomly at the beginning of the experiment and the same animals were used for all xenodiagnoses trials.

At the end of the experiments (20 weeks p.i.) the rodents were euthanised, dissected, and tissues from ears, paws, tail, ear-draining lymph nodes, spleen, and liver were stored at − 20°C for subsequent analyses of parasite load (DNA isolation, nested PCR and qPCR).

## 2.4. Morphometry of parasites

The inocula were fixed with methanol, stained with Giemsa, and examined under the light microscope with an oil-immersion objective. One hundred and thirty randomly selected promastigotes from each infection dose were photographed using the QuickPHOTO MICRO programme for further analysis in the ImageJ programme. The body and flagellar lengths of the parasites were measured and metacyclic forms were distinguished, based on the criteria of Walters (1993) and Cihakova and Volf (1997) [40,41]: body length < 14 μm and flagellar length 2 times body length.

## 2.5. DNA isolation and nested PCR of sand fly and rodent samples

Total DNA isolated using the High Pure PCR Template Preparation Kit (Roche Diagnostics, Indianapolis, IN) according to the manufacturer's instructions was used as a template for nested PCR amplification (detection limit $10^{-2}$ parasites per sample) with outer primers amplifying 332 bp region of 18S sequence (forward primer 18SN1F 5'- GGA TAA CAA AGG AGC AGC CTC TA3' and reverse primer 18SN1R 5'–CTC CAC ACT TTG GTT CTT GAT TGA-3') and inner primers amplifying the 226 bp long 18S sequence (forward primer 18SN2F 5′-AGA TTA TGG AGC TGT GCG ACA A-3′ and reverse primer 18SN2R 5′-TAG TTC GTC TTG GTG CGG TC-3′) previously used by Sadlova et al. 2022 [42]. The samples were then analysed using 0,8% agarose gel. The reaction mixtures and cycling conditions were as follows:

1. step of PCR: 3 μl of genomic DNA, 0.5 μl forward primer 18SN1F (10 μM), 0.5 μl reverse primer 18SN1R (10 μM), 10 μl of 2x EmeraldAmp GT PCR Master Mix (Takara Bio), 6 μl of ddH2O. Step 1, 94°C for 3 min 30 s; step 2, 94°C for 30 s; step 3, 60°C for 30 s; step 4, 72°C for 25 s; step 5, 72°C for 7 min; followed by cooling at 12°C. Steps 2–4 were repeated 35 times.

2. step of PCR: 1 μl of 1. step PCR reaction, 0.5 μl forward primer 18SN2F (10 μM), 0.5 μl reverse primer 18SN2R (10 μM), 10 μl 2x EmeraldAmp GT PCR Master Mix (Takara Bio), 8 μl

ddH2O. Step 1, 94°C for 30 s; step 2, 94°C for 30 s; step 3, 60°C for 30 s; step 4, 72°C for 20 s; step 5, 72°C for 7 min; followed by cooling at 12°C. Steps 2–4 were repeated 35 times.

### 2.6. Quantitative PCR

Parasite quantification by quantitative PCR (qPCR) was performed in a Bio-Rad iCycler & iQ Real-Time PCR Systems using the SYBR Green detection method (SsoAdvanced Universal SYBR Green Supermix, Bio-Rad, Hercules, CA). Primers targeting the 226 bp long 18S sequence (forward primer 18SN2F 5′-AGA TTA TGG AGC TGT GCG ACA A-3′ and reverse primer 18SN2R 5′-TAG TTC GTC TTG GTG CGG TC-3′) were used. One microlitre of DNA was used per individual reaction. PCR amplifications were performed in duplicates using the following conditions: 98°C for 2:30 min followed by 40 repetitive cycles: 98°C for 10 s and 60°C for 20 s. PCR water was used as a negative control. Detection limit of used assay is $10^3$ parasites per sample. A series of 10-fold dilutions of *L. martiniquensis* promastigote DNA, ranging from $5 \times 10^6$ to $5 \times 10^1$ parasites per PCR reaction, was used to prepare a standard curve. Quantitative results were expressed by interpolation with a standard curve. To monitor non-specific products or primer dimers, a melting analysis was performed from 70 to 95°C at the end of each run, with a slope of 0.5°C/c, and 6 s at each temperature. The parasite loads in tested tissues were classified into three categories: low, < 1000 parasites; moderate, 1000–10 000 parasites; heavy, > 10 000 parasites.

### 2.7. Statistical analysis

Differences in infection rates were analysed using Fishers Exact Tests in SPSS version 27.

## 3. Results

### 3.1. Development of *Mundinia* in BALB/c mice

Two independent trials, each with groups of 5 animals inoculated with culture-derived *Mundinia* species, were processed; in total, 10 BALB/c mice infected with each *Mundinia* species (*L. enriettii*, *L. macropodum*, *L. orientalis*, *L. chancei*, and four strains of *L. martiniquensis)* were studied. The representation of the infective metacyclic stages in the inocula ranged between 0–32% (S1 Table).

 Mice did not show weight loss or external signs of infection throughout the experiment (S2 Table). Xenodiagnoses were performed at weeks 5, 10, 15 and 20 p.i. on 4/10 mice from each group and revealed no positive results by PCR. Also, all examined tissue samples (i.e., ears, paws, tail, ear-draining lymph nodes, spleen and liver) were negative for the presence of *Leishmania* DNA by PCR (S3 Table). In summary, BALB/c mice were resistant to all *Mundinia* species tested.

### 3.2. Development of *Mundinia* in Chinese hamsters (*Cricetulus griseus*)

In total, 10 Chinese hamsters were infected with each of the culture-derived *Mundinia* tested: *L. enriettii*, *L. macropodum*, *L. orientalis*, *L. chancei*, and four strains of *L. martiniquensis*. In total, 80 animals were used. The representation of infectious metacyclic stages in the inocula ranged between 0–57% (S1 Table)

 Similar to BALB/c mice, Chinese hamsters did not show external signs of infection or weight loss (S2 Table) during the experiment. However, hamsters infected with *L. enriettii*, *L. chancei*, *L. orientalis*, and *L. martiniquensis* were infectious to sand flies throughout the experiment from week 5 p.i. to week 15 p.i. in *L. enriettii* and to week 20 p.i. in *L. chancei*, *L. orientalis*, and *L. martiniquensis*, when the experiment was terminated (Table 1). Always, 4/10

**Table 1. Infectiousness of tested Chinese hamsters to sand flies during xenodiagnoses.**

|  | 5 weeks p.i. | 10 weeks p.i. | 15 weeks p.i. | 20 weeks p.i. |
|---|---|---|---|---|
| *L. enriettii* | 1/9 | 0/17 | 1/8 | 0/8 |
| *L. macropodum* | 0/7 | 0/13 | 0/6 | 0/12 |
| *L. chancei* | 1/9 | 0/14 | 1/9 | 1/13 |
| *L. orientalis* | 6/11 | 2/16 | 0/12 | 1/12 |
| *L. martiniquensis* Mar1 | 0/7 | 0/14 | 1/18 | 0/7 |
| *L. martiniquensis* Cu1R1 | 1/9 | 0/11 | 0/5 | 0/8 |
| *L. martiniquensis* Cu2 | 1/11 | 0/16 | 1/11 | 1/13 |
| *L. martiniquensis* Aig1 | 0/11 | 0/12 | 0/7 | 0/8 |

The left numbers represent the number of PCR positive pools and the right numbers represent the total number of tested pools. Each pool consisted of 5 engorged *Phlebotomus duboscqi* females.

animals per group were used for xenodiagnoses trials. Nested PCR showed positivity of the inoculated ears in 6/8 tested groups at the end of the experiment, only samples from animals infected with *L. macropodum* and *L. martiniquensis* Aig1 were negative. *Leishmania enriettii* DNA was found in the inoculated ear of one animal and also in draining lymph nodes of the inoculated ear of another animal. The inoculated ears of animals infected with *L. chancei* were positive in 30% and parasites were not detected in any other tissue. In hamsters infected with *L. orientalis*, 40% of the ears were positive by nested PCR, while in one case the parasites were detected in the draining lymph nodes of the inoculated ear. Animals infected with *L. martiniquensis* MAR1 showed 20% of inoculated ears and draining lymph nodes positive for parasite DNA. In animals infected with *L. martiniquensis* Cu1R1, 30% of the inoculated ears were positive by nested PCR while no parasite DNA was detected in any other tissue. The highest proportion of infected animals was found in the group infected with *L. martiniquensis* Cu2 where 80% of the inoculated ears were positive and parasite DNA was also detected in 10% of the hind paws (Fig 1). Quantitative PCR showed that only a low numbers of parasites (<1000) were present in all the tissues tested (S3 Table).

### 3.3. Development of *Mundinia* in steppe lemmings (*Lagurus lagurus*)

As with the previous host species, groups of 5 steppe lemmings were inoculated with five culture derived species of *Mundinia* (*L. enriettii*, *L. macropodum*, *L. orientalis*, *L. chancei*, and four strains of *L. martiniquensis*) and the experiment was repeated twice; in total, 10 individuals of steppe lemmings were infected by each parasite species. The proportion of metacyclic stages in inocula ranged between 0–25% (S1 Table).

In steppe lemmings, most *Mundinia* species and strains produced significantly higher infection rates than in Chinese hamsters (Figs 2 and 3). The pathological signs of infection (Figs 4 and 5) including weight changes (S2 Table) in steppe lemmings varied from asymptomatic through cutaneous to visceral form. However, specimens in all groups were infectious to sand flies from week 5 p.i. until the end of the experiment by week 20 p.i. (Table 2).

In all animals infected with *L. enriettii*, parasite DNA was detected in the inoculated ears and also in 30% of the draining lymph nodes of the inoculated ears. Parasite DNA was also present in 10% of the forepaws, hind paws, tails, and spleens (Fig 3). Although the number of parasites in infected ears was low or moderate (maximum $2.4 \times 10^4$), a very high proportion of pools of sand flies used for xenodiagnoses were positive (more than 85% from week 10 p.i.) (Table 2). All animals that were used for xenodiagnoses developed cutaneous symptoms (swelling and dry lesions) on the ears from week 8 or 9 p.i., respectively, which persisted to the

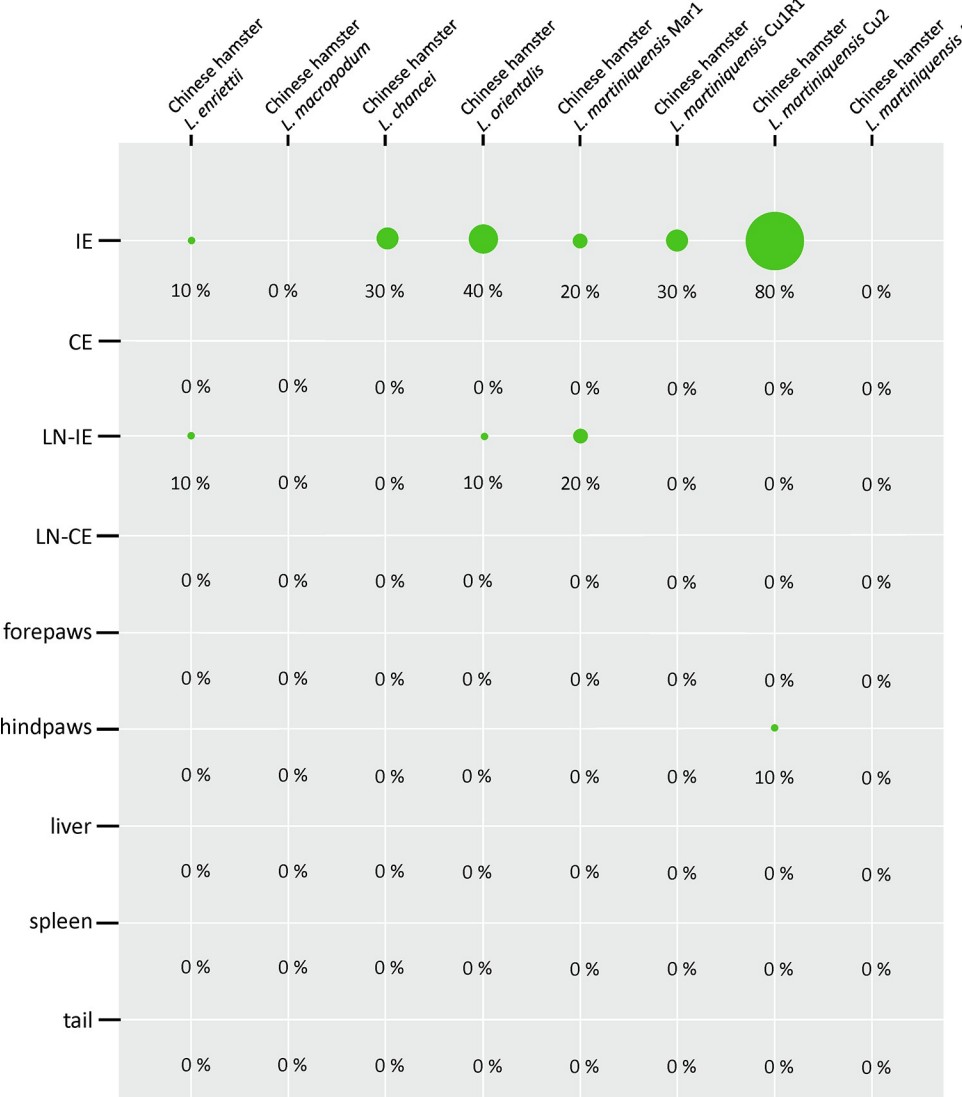

**Fig 1. Anatomical distribution of parasites in Chinese hamsters determined by nested PCR.** The results are presented by the balloon graph where the size corresponds to the percentage of infected tissues from the total sum of tested organs of the same type. IE = inoculated ears; CE = contralateral ears; LN-IE = draining lymph nodes of inoculated ears; LN-CE = draining lymph nodes of the contralateral ears.

end of the experiment while other animals did not show any cutaneous symptoms. In one animal, which was not used for xenodiagnoses, we also observed hair loss in the abdomen from the 17th week p.i. (Fig 3).

Animals infected with *L. macropodum* did not show any symptoms of infection during the experiment. We detected only low numbers of parasites (<1000) in 33% of inoculated ears (again, only in animals used for xenodiagnoses) and in the tail of one additional animal not exposed to sand flies (Fig 3). Steppe lemmings were infectious to sand flies only in 11% by week 5 p.i. and 33% by week 20 p.i. (Table 2).

In steppe lemmings infected with *L. chancei*, one animal died 2 weeks post infection from unknown reasons. The development of the lesion was observed in only one animal used for the xenodiagnosis. The lesion with swelling began to form by week 8, leading to partial

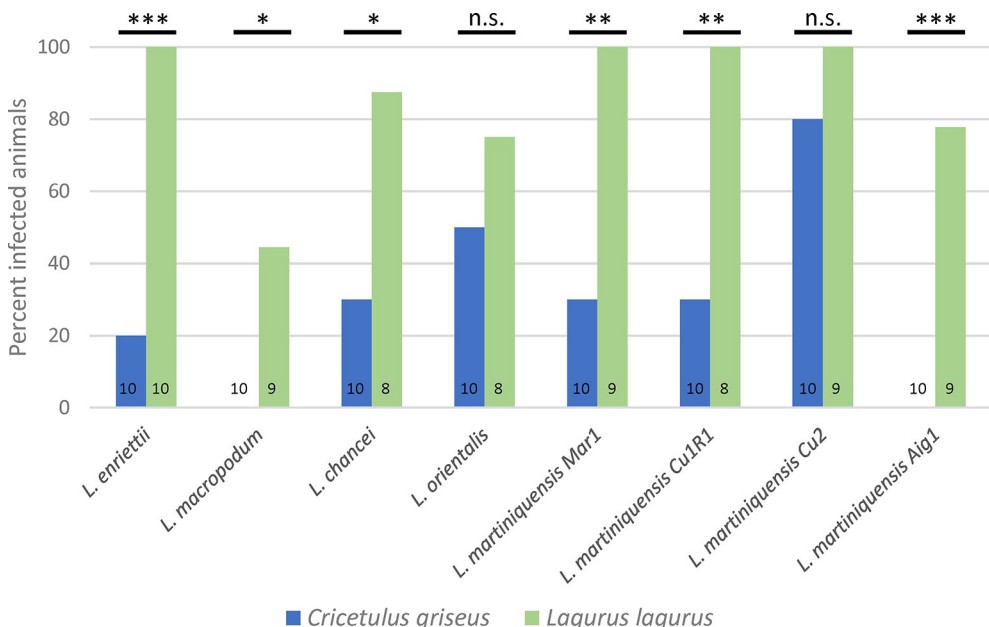

**Fig 2. Infection rates in Chinese hamsters (*Cricetulus griseus*) and steppe lemmings (*Lagurus lagurus*) infected with *Mundinia*.** The numbers of animals evaluated are shown in the graph. P values indicate difference between host species; n.s., nonsignificant difference, *, P < 0.05, **, P < 0.01, ***, P < 0.0001.

destruction of the pinna by week 13 (Fig 4C and 4D). The loss of hair in the abdomen appeared in two specimens between 10 and 12 weeks (Fig 4F) and one of these animals died 3 weeks later. All other 6 animals showed no external signs of infection during the entire experiment. However, parasites disseminated throughout the body and were found in all tested tissues except lymphatic nodes draining non-inoculated ears, although one of the tested non-inoculated ears tested was positive for the presence of parasites (Fig 3). The positivity rate of inoculated ears, lymph nodes that drain them, hind paws and spleen was 37.5%. Although the positivity and infectiousness of the animals were high, only low or moderate numbers of parasites (<2500 per tested tissue) were detected by qPCR (S3 Table).

Among the animals infected with *L. orientalis*, 4/10 specimens died during the experiment, by weeks 7, 8, 13 and 15 p.i. One specimen that died by week 13 showed hair loss and weight loss, while no external signs of infection were recorded in others. Nested PCR revealed that 75% of the animals were positive for *Leishmania* DNA at the time of dissection. Parasites spread into all tested tissues except the tail (Fig 3), most frequently they were present in the lymph nodes (75%), forepaws (62.5%), spleen (50%) and liver (37.5%). Similarly to animals infected with *L. chancei*, the positivity and infectiousness of the animals were high, but parasite numbers were only low or moderate (<2500) (S3 Table).

Large differences were observed in animals infected with various strains of *L. martiniquensis*. Although *L. martiniquensis* strains Mar1, Cu1R1, and Aig1 did not cause any pathological signs of infection, animals infected with *L. martiniquensis* Cu2 showed the most severe cutaneous signs of infection of all tested groups. All four animals that were used for xenodiagnoses developed pathological changes in the pinnae, while none of the five animals that were not used for xenodiagnoses developed lesions. After the first round of xenodiagnoses 5 weeks p.i., swelling and lesions began to appear (Fig 4B) and continued to grow in two animals, leading to the destruction of the pinna by week 12 (Fig 4E). In another animal, the lesion started to form by week 12 and the pinna was completely destructed by week 19, and one animal lost the

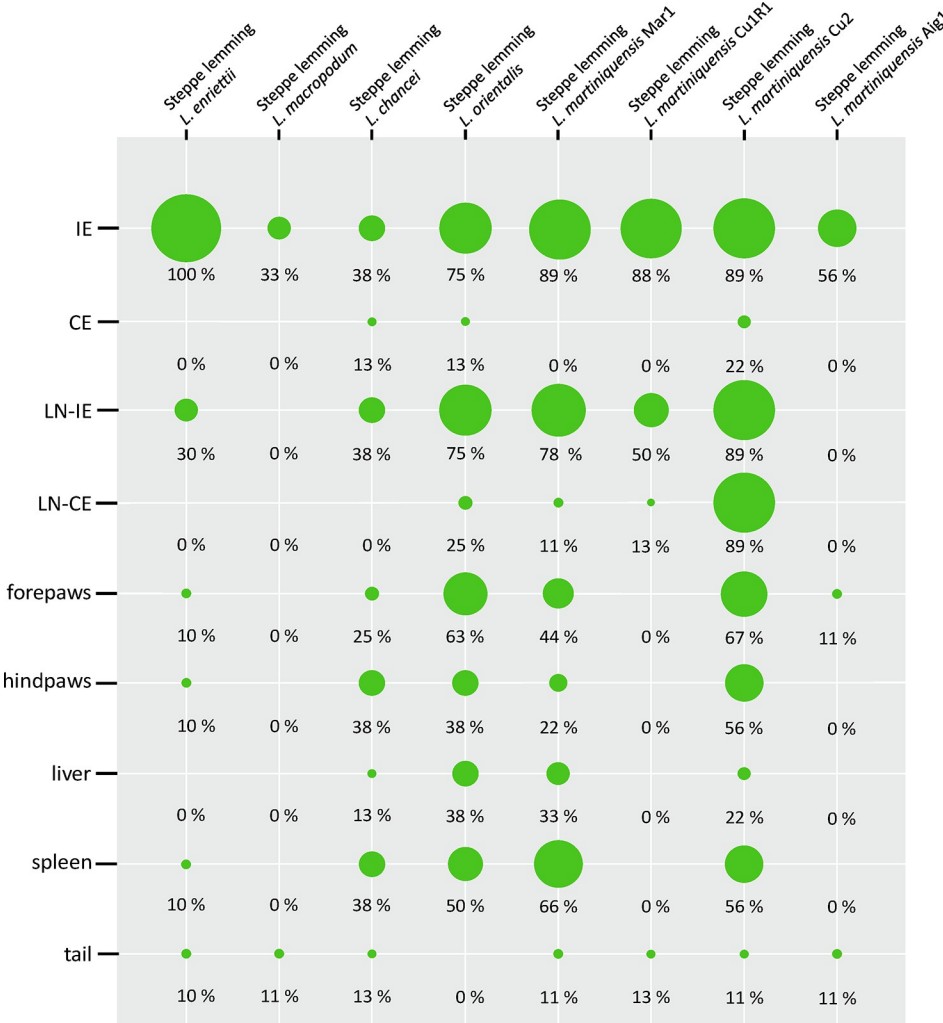

**Fig 3. Anatomical distribution of parasites in steppe lemmings determined by nested PCR.** The results are presented by the balloon graph where the size corresponds to the percentage of infected tissues from the total sum of tested organs of the same type. IE = inoculated ears; CE = contralateral ears; LN-IE = draining lymph nodes of inoculated ears; LN-CE = draining lymph nodes of the contralateral ears.

entire pinna between weeks 16 to 19, while no pathological signs were observed before that (Fig 5). One animal did not show any lesions, but weight loss was observed from week 10 and its overall health status degraded until the end of the experiment. Four other animals (that have not been xenodiagnosed) showed no external signs of infection and one animal died 6 weeks p.i. Parasites were detected by PCR in 88.9% (8/9) of the animals. The parasites often spread into all tissues tested (Fig 3), most frequently to lymph nodes (88.9%), forepaws (66.7%), hind paws (55.6%), and spleen (55.6%) (Fig 3). Quantitative PCR revealed only low numbers of *Leishmania* in most tissues tested, but higher numbers were present in 6/9 inoculated ears ($7 \times 10^3$–$3.16 \times 10^5$) and $4 \times 10^4$ parasites were detected in the spleen of the animal with the highest parasite load in the inoculated ear (S3 Table). Interestingly, this specimen did not show signs of infection (S2 Table). Animals infected with *L. martiniquensis* Mar1 were infectious to sand flies from week 5 p.i. to the end of experiment (Table 2). Parasite DNA was found in all tested tissues except of contralateral ears, most often in inoculated ears (89%),

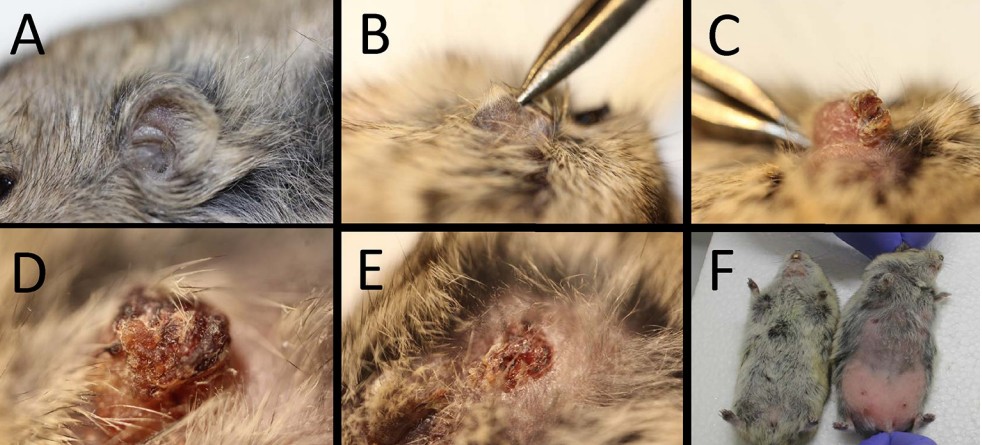

**Fig 4. Pathological changes caused by *Mundinia* in the steppe lemming (*Lagurus lagurus*).** A) healthy ear before inoculation; B) lesion formation in animal infected with *L. martiniquensis* Cu2 12 weeks p.i.; C) dry lesion in animal infected with *L. martiniquensis* Cu2 15 weeks p.i.; D) partial pinna necrosis in animal infected with *L. chancei* 13 weeks p.i.; E) destruction of the ear and dissemination of parasites around the eye in animal infected with *L. martiniquensis* Cu2 19 weeks p.i.; F) hair loss in abdomen of animal infected with *L. chancei* 12 weeks p.i. (animal on the right).

draining lymph nodes of inoculated ears (78%), spleen (66%) and forepaws (44%) (Fig 3). Low or moderate numbers of parasites were detected in inoculated ears (maximum $3 \times 10^4$) and only low numbers of parasites (<1000) were detected in other tested tissues (S3 Table). In animals infected with *L. martiniquensis* Cu1R1, parasites were detected in inoculated ears (88%), draining lymph nodes of inoculated ears (50%), draining lymph nodes of contralateral ears and tail (13%) (Fig 3). The animals were infectious to sand flies throughout the course of experiment, but only in a low proportion compared to other strains of *L. martiniquensis* with positive pools ratio slowly increasing over time (Table 2). In steppe lemmings infected with the Aig1 strain, parasites were found in 56% of inoculated ears and 11% of forepaws and tails (Fig 3). Although only low numbers of parasites were detected (S3 Table), the xenodiagnoses proved that animals were infectious to sand flies, even in 50% by week 20 p.i. (Table 2).

## 4. Discussion

Leishmaniases caused by *Mundinia* species are an emerging health problem that cannot be ignored. While cases of animal leishmaniasis caused by *L. enriettii*, *L. macropodum*, and *L. procaviensis* have been described only sporadically, human disease caused by *L. martiniquensis* and *L. orientalis* is on the rise in Southeast Asia, especially in Thailand [9,43], and cutaneous leishmaniasis caused by *L. chancei* is emerging in Ghana [15,25]. Since the knowledge of natural reservoirs is limited, it is necessary to develop a reliable model for experimental research on the biology of these parasites.

Among the classical laboratory models, guinea pigs (*Cavia porcellus*) were first used for *Mundinia* research. These animals have been shown to be susceptible to *L. enriettii* [9,44–47], however, infections with other *Mundinia* species were lost; only guinea pigs infected with *L. orientalis* and *L. martiniquensis* showed temporary pathological changes in the ear pinnae [47]. In golden hamsters (*Mesocricetus auratus*), *L. enriettii* showed only mild symptoms with temporary lesions at the site of inoculation and their subsequent healing [9,43] while *L. martiniquensis* disseminated, causing signs of VL [48]. Recently, *L. chancei* was tested in *Mastomys natalensis* and *Arvicanthis niloticus* as these two rodents inhabit Ghana and could potentially

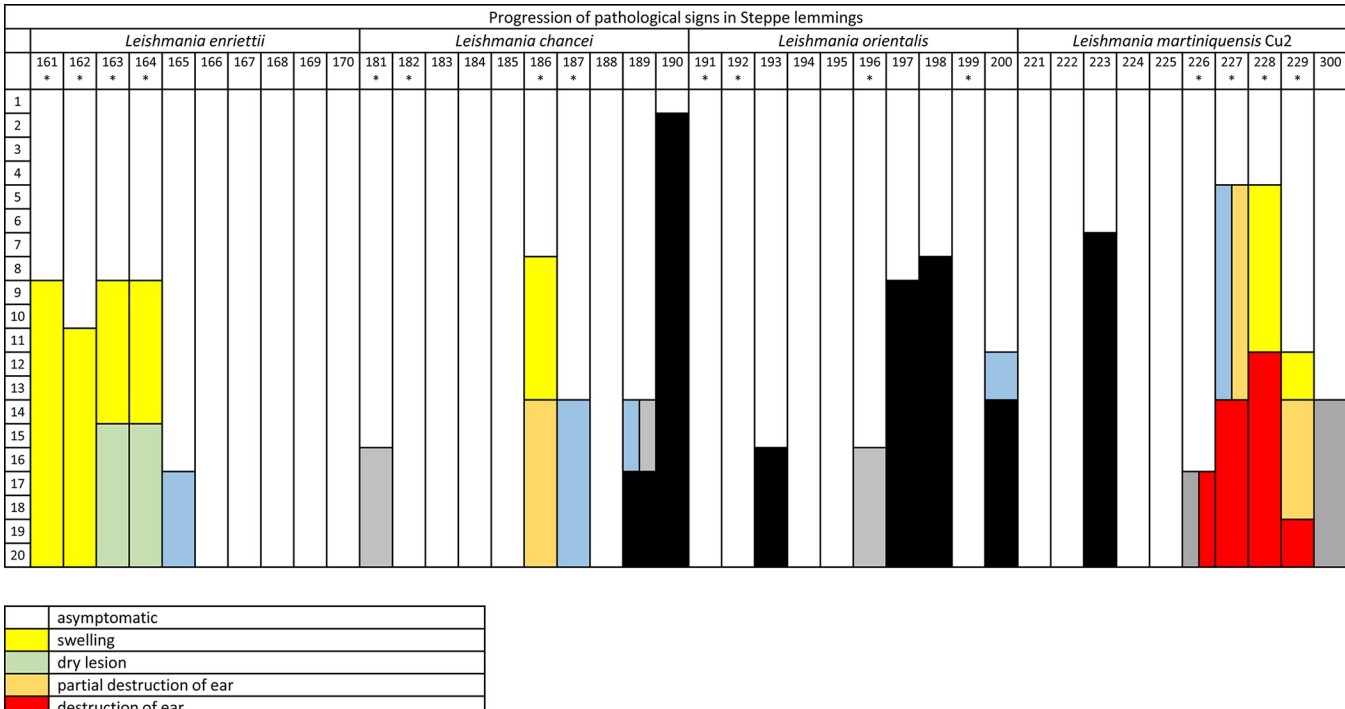

**Fig 5. Graphical table showing pathological changes observed in steppe lemmings (*Lagurus lagurus*) infected with *Mundinia*.** The categories were: asymptomatic (white); swelling (yellow); dry lesion (green); partial destruction of the pinna (orange); destruction of the pinna (red); abdomen hair loss (blue); weight loss (grey); death (black). Axis Y describes week post infection, and axis X shows the number codes of examined animals (each column represents one animal). Specimens used for xenodiagnoses trials are marked with * sign.

serve as reservoir hosts of the parasite. However, *Leishmania* survival in both hosts was limited, making them unsuitable experimental models [36].

In the present study, we tested three rodent species: BALB/c mouse (*M. musculus*), Chinese hamster (*C. griseus*), and steppe lemming (*L. lagurus*). BALB/c mice were chosen as the most commonly studied murine model, highly susceptible to cutaneous leishmaniasis caused by *L. major* as well as visceral leishmaniasis (reviewed by [49]). Chinese hamsters and steppe

**Table 2. Infectiousness of tested steppe lemmings to sand flies during xenodiagnoses.**

|  | 5 weeks p.i. | 10 weeks p.i. | 15 weeks p.i. | 20 weeks p.i. |
|---|---|---|---|---|
| *L. enriettii* | 6/15 | 11/13 | 14/15 | 16/18 |
| *L. macropodum* | 1/13 | 0/13 | 0/9 | 3/9 |
| *L. chancei* | 2/12 | 0/9 | 1/10 | 2/5 |
| *L. orientalis* | 5/10 | 0/6 | 3/7 | 3/7 |
| *L. martiniquensis* Mar1 | 8/14 | 4/13 | 7/13 | 4/12 |
| *L. martiniquensis* Cu1R1 | 2/14 | 1/12 | 2/15 | 3/10 |
| *L. martiniquensis* Cu2 | 3/4 | 2/5 | 5/8 | 5/8 |
| *L. martiniquensis* Aig1 | 2/12 | 0/14 | 9/17 | 4/8 |

The left numbers represent the number of PCR positive pools and the right numbers represent the total number of tested pools. Each pool consisted of 5 engorged *Phlebotomus duboscqi* females.

lemmings were used as these animals have been shown to be more susceptible to *L. donovani* than golden hamsters, the most common rodent model for visceral leishmaniasis: In experiments where the three rodent species were directly compared after intradermal inoculation of *L. donovani*, both Chinese hamsters and steppe lemmings showed more extensive spread of parasites over the body and higher parasite loads and infectiousness to sand flies then golden hamsters. In addition, both Chinese hamsters and steppe lemmings were also highly susceptible to cutaneous *L. major* [33]. We inoculated culture-derived parasites intradermally into rodent pinnae to simulate the natural mode of infection to the best of our knowledge. Indeed, the use of vector-derived parasites together with homogenate of salivary glands would be a more appropriate way to simulate natural mode of transmission [50], however, this experimental scheme was not applicable in the study since the natural vectors of these parasites are unknown.

BALB/c mice were proved to be resistant to infection with all five *Mundinia* species tested. At the end of the experiment, by week 20 p.i., *Leishmania* DNA was not detected in any of the tissues tested, and the rodents were not infectious to sand flies throughout the whole experiment. Our results were consistent with previous observations: BALB/c mice infected with *L. martiniquensis* strains MAR1 and MAR2 were asymptomatic and no mortality, weight loss, or clinical signs were observed in mice infected with these strains, regardless of the route of inoculation. But parasite dissemination and infection kinetics varied by inoculation method and inoculum size [51]. Somboonpoonpol (2016) [52] described the poor development of Thai *L. martiniquensis* after subcutaneous inoculation whereas high parasite burdens were observed after i.v. application of the parasite. Intakhan et al. (2020) [48] used intraperitoneal infection and detected parasite DNA only in the liver and spleen of infected animals in low numbers at 8 weeks p.i. followed by clearance at week 16 p.i. in most animals. These results suggest that BALB/c mice are not a suitable laboratory model to study the pathology of *Mundinia* infection at least after intradermal inoculation. However, there are other inbred mouse models that differ in their immune response to leishmaniasis from the BALB/c strain. It may be helpful to study the development of *Mundinia*, for example, in C57BL/6 mice, which, unlike the BALB/c strain, are resistant to *L. major* but are susceptible to *L. amazonensis* or *L. mexicana* (reviewed by [4]). Another option is to study *Mundinia* in outbred mouse strains that demonstrated an intermediate phenotype between resistant and susceptible inbred strains [53].

Chinese hamsters (*C. griseus*) were demonstrated as susceptible to *L. enriettii*, *L. chancei*, *L. orientalis* and three strains of *L. martiniquensis* (MAR1, Cu1R1, and Cu2), even though no external signs of disease were observed. All hamsters were in perfect health until the end of the experiment, a low number of parasites were localized in their inoculated ears, and the animals were sporadically infectious to sand flies. Very rarely, parasites were detectable in lymph nodes draining the inoculated ear and hind paws, but also in low numbers. These data suggest that Chinese hamsters have the potential to mimic asymptomatic infection. These asymptomatic infections seem to play a major role in the circulation of *Mundinia*, as no symptomatic wild hosts have yet been found in nature [5] and most human infections in Thailand are also asymptomatic [54]. On the other hand, except for *L. martiniquensis* Cu2 (80% of positive animals) and *L. orientalis* (50% of positive animals), the proportion of animals with detectable parasites was very low, which would be problematic for more robust and detailed studies.

Steppe lemmings (*Lagurus lagurus*) were shown to be highly susceptible to all *Mundinia* species tested, with severe signs of infection observed throughout the experiment in human infecting species. Animals were infectious to sand flies and parasites were detected in all tissues tested. Interestingly, only lemmings used for xenodiagnoses developed pathological signs on the ears. This is consistent with previously stated hypothesis of enhancing effect of sand fly bites to the course of infection by Vojtkova et al. (2021), who observed significantly faster and

to a larger size lesion development in *L. major* infected BALB/c mice exposed to repeated bites of *P. dubosqci* compared to the unexposed group of mice, and the tissue parasite load was higher in rodents exposed to repeated bites [55]. Also in hamster model, increased transmissibility of *L. donovani* was observed following multiple exposures of hamsters to *Lutzomyia longipalpis* bites [56].

The present study demonstrated that steppe lemmings support the survival and development of *L. enriettii*, which caused small dry skin lesions in animals exposed to sand flies, while other animals, although positive for parasites at the end of the experiment, remained asymptomatic. Interestingly, a very high percentage of positive sand fly pools (over 90%) from xenodiagnoses was observed, although we found only a low or moderate number of parasites in the inoculated ears. Also, unlike guinea pigs, where lesions heal and reduction of infectiousness to sand flies with time was observed [47], steppe lemmings did not heal lesions over time and animals remained infectious to sand flies until the end of the experiment which brings the potential to study the non-healing phenotype of cutaneous leishmaniasis and supports our hypothesis that in nature, animals infected with *Mundinia* show mild signs of infection or remain asymptomatic and may serve as reservoirs for a long term.

Up to date, it was thought that *L. macropodum* is restricted to kangaroos and circulate in nature in these marsupials being transmitted by biting midges [7,12]. Here we prove these parasites can infect, survive and be infectious to vectors even in rodents. Steppe lemmings were infectious to sand flies in 11% by week 5 p.i., then any of the tested pools was positive for presence of parasites for 10 weeks, but even 33% of pools were positive by week 20 p.i. By this time, parasites were present in inoculated ears and rarely in the tail. Due to these findings, we hypothesise parasites can develop well in ears for limited period of time, then they are suppressed by immune system, but they can survive, slowly overcome immunological barriers, and start to multiply again. It would be interesting to perform long term experiments and observe dynamics of the infection in several time intervals.

Interestingly, only animals exposed to sand flies remained infected, whereas no parasites were observed in other animals, which also supports the hypothesis of sand fly bites influence on the course of infection.

Animals infected with *L. chancei* showed a mixture of cutaneous and visceral symptoms, with 7/8 animals positive in various tissues and organs. On the other hand, the xenodiagnoses were positive only sporadically. The visceralisation observed in steppe lemmings is unparalleled to human infections described in Ghana and neighbouring countries. Here, among 8000 cases of leishmaniasis none of the cases was described as visceral, only skin symptoms were observed [15]. However, the results of current study suggest that *L. chancei* could have the potential to visceralise, for example, in immunocompromised individuals such as HIV-positive people.

*Leishmania chancei* is most closely related to *L. orientalis* from Thailand [23], where variable human disease manifestations correlated with patients' immunocompetence status can be observed. The type strain LSCM4 [22] used also in our study and the strain CULE5 [23] caused simple skin lesions in immunocompetent patients while in AIDS patients, the strain PCM2 Trang and the anonymous isolate from Kanchanaburi caused disseminated cutaneous leishmaniasis and visceral disease [16,21]. Steppe lemmings infected with *L. orientalis* displayed no skin lesions, but 4/10 of the animals died during the experiment and one more suffered from hair loss, which is a usual symptom of visceral leishmaniasis similar to their spread widely into various body tissues and organs (38% of liver and 50% of spleen were positive by PCR). Previously, a high susceptibility of steppe lemmings to progressive VL caused by *L. donovani* has been reported [33,34], so the present study extends the spectrum of *Leishmania* species for which steppe lemmings may be a suitable animal model for future studies.

This study presents a unique comparison of four strains of *L. martiniquensis* originating from various continents and hosts. While the strains Cu1R1 and Cu2 came from humans living in Thailand, Mar1 originates from human in Martinique and Aig1 was isolated from a horse in the Czech Republic. *Leishmania martiniquensis* is probably the most widespread species of *Leishmania*, which was hidden from scientists until the 1990s and causes various symptoms in different hosts. While the Cu2 strain was destructive to ear pinnae of animals and was present in moderate or high numbers, other strains did not cause any symptoms for the whole course of experiment and they were present only in low numbers in tissues. Interestingly, strains Cu1R1 and Cu2 coming from the same area developed differently in steppe lemmings: while Mar1 and Cu2 visceralised, strains Cu1R1 and Aig1 remained mainly in inoculated ears or their draining lymph nodes. These differences point to polymorphisms between strains within the species similar to those observed in *L. major* [57].

*Mundinia* is remarkable, important and poorly described subgenus of the genus *Leishmania*. Its worldwide distribution, wide spectrum of hosts, various clinical manifestations, and probable involvement of unusual vectors add more mystery to the whole story. Unravelling questions concerning vectors, reservoir hosts, etc. is a prerequisite for describing the life cycle of these parasites. The use of new animal models, such as steppe lemmings and Chinese hamsters, can contribute significantly to this goal. In general, wild rodents represent a novel approach to the study of host competence and host-parasite relationships [4]. The use of only one inbred genotype may not correspond to the natural variability in parasite's tissue tropism, the dynamics and duration of infection or the infectiousness to vectors. Since the manifestations of human diseases caused by *Mundinia* are variable and show almost as polymorphic disease course as those caused by *Leishmania* and *Viannia* subspecies, it would be optimal to introduce a wider range of model animals. Conversely, for studies aimed at drug testing or the development of diagnostic methods, it would be preferable to use less genetically variable models, either obtained by long-term inbreeding of these rodent species or by searching among existing inbred mouse models different from BALB/c. The results of this work must be considered pioneering and indicative of the direction of future research.

## Supporting information

**S1 Table. Representation of metacyclic promastigotes in inocula.**
(XLSX)

**S2 Table. Weight changes of infected animals during the experiment.**
(XLSX)

**S3 Table. Parasite load results from various tissues measured by qPCR.**
(XLSX)

## Acknowledgments

We would like to thank prof. Padet Siriyasatien, prof. Paul Bates and doc. Jan Votýpka and prof. David Modrý for providing *Leishmania* isolates. Also, we would like to thank our technical support stuff Kristyna Srstkova, Lenka Krejcirikova, and Lenka Hlubinkova.

## Author Contributions

**Conceptualization:** Tomas Becvar, Petr Volf, Jovana Sadlova.

**Data curation:** Tomas Becvar.

**Formal analysis:** Tomas Becvar.

**Funding acquisition:** Tomas Becvar, Petr Volf.

**Investigation:** Tomas Becvar, Barbora Vojtkova, Lenka Pacakova, Barbora Vomackova Kykalova, Lucie Ticha, Jovana Sadlova.

**Methodology:** Tomas Becvar, Petr Volf, Jovana Sadlova.

**Resources:** Petr Volf.

**Supervision:** Petr Volf, Jovana Sadlova.

**Validation:** Tomas Becvar.

**Writing – original draft:** Tomas Becvar.

**Writing – review & editing:** Petr Volf, Jovana Sadlova.

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
