## [Decision Letter · Decision Letter 0]

10 Mar 2024

Dear doc. Sádlová,

Thank you very much for submitting your manuscript "Steppe lemmings and Chinese hamsters as new potential animal models for the study of the leishmania subgenus <i>Mundinia<i/> (Kinetoplastida: Trypanosomatidae)" for consideration at PLOS Neglected Tropical Diseases. As with all papers reviewed by the journal, your manuscript was reviewed by members of the editorial board and by several independent reviewers. In light of the reviews (below this email), we would like to invite the resubmission of a significantly-revised version that takes into account the reviewers' comments. 

We cannot make any decision about publication until we have seen the revised manuscript and your response to the reviewers' comments. Your revised manuscript is also likely to be sent to reviewers for further evaluation.

Sincerely,

Nadira D. Karunaweera

Academic Editor

Abhay Satoskar

Section Editor

Reviewer's Responses to Questions

**Key Review Criteria Required for Acceptance?**

**Methods**

-Are the objectives of the study clearly articulated with a clear testable hypothesis stated?

-Is the study design appropriate to address the stated objectives?

-Is the population clearly described and appropriate for the hypothesis being tested?

-Is the sample size sufficient to ensure adequate power to address the hypothesis being tested?

-Were correct statistical analysis used to support conclusions?

-Are there concerns about ethical or regulatory requirements being met?

Reviewer #1: (No Response)

Reviewer #2: This study presents the experimental evidence of the long-term survival of Mundinia parasites in new potential animal models as well as their infectiousness to vectors. Thus, the research content is interesting and worth publishing but needs correction and more critical discussion.

Major comments:

1. The authors mentioned Culicoides as the most probable vectors of Mundinia species with the previous evidence of successful metacyclogenesis and transmission to animal model. In addtition, it appears that transmission of human-infecting Mundinia species entirely failed with Phlebotomus argentipes. However, Phlebotomus, not Culicoides, was used for xenodiagnosis in the present study. What is the rationale of using Phlebotomus for xenodiagnosis in this research? Additonally, xenodiagnosis was not mentioned in the objective. So, the authors should include the rationale and objective regarding this in the introduction.

2. Line 80: Only four cases of L. orientalis has formally been reported. Please include two more relevant references as follows:

doi: 10.1186/s13071-018-2908-3 and doi: 10.4269/ajtmh.22-0385.

3. Line 90: What are "a sink" and "a source"? To facilitate the general readers, please clarify these two terms.

4. Figure 2 shows only the anatomical distribution of parasites in steppe lemmings. Do the authors have the similar figure for those in chinese hamsters? 

5. The authors proved the organ infection by only nested-PCR. Do the authors have the histological validation? 

6. Lines 372-374: How do the sand fly bites influence the courese of infection? Please describe more.

7. Lines 375-376: As recorded, L. chancei (previously known L. sp. Ghana) has been known to cause cutaneous leishmaniasis in Ghana. Visceral leishmaniasis has never been epidemiologically reported before for this species. Thus, this finding of cutaneous and visceral symptoms in this study unveil the possible risks of visceral involvement in humans by this species. I suggest the authors to include this issue in this discussion.

8. Lines 377-388: Surprisingly, L. orientalis displayed no cutaneous lesions but caused visceral symptoms resulting in deaths in steppe lemmings. Conversely, in humans this Mundinia species has previously been known to be mainly responsible for cutaneous leishmaniasis. Please discuss more. 

9. What is the limitations and implications/implementation (in terms of diagnostics and therapeutic development) in this study? The authors should critically discuss more.

Mino comments:

1. Lines 73-75: Please recheck the format of the reference.

2. The term "leishmania" which is a genus name need to be itaic and start with the capital letter "L". Please correct throughout the manuscript.

Reviewer #3: The hypothesis and objectives of the study were clearly stated and the design of study was appropriate. To fulfil requirements for animal model statistics and ethics, the number of animals used which were a group of 10 animals for infection of each species was appropriate and suitable. Fisher's exact test was used for statiscal analysis to compare infection rates which is suitable for animal infection model studies. Ethics approval numbers have been documented in the manucscript to prove ethical and regulatory requirements have been met.

**Results**

-Does the analysis presented match the analysis plan?

-Are the results clearly and completely presented?

-Are the figures (Tables, Images) of sufficient quality for clarity?

Reviewer #1: (No Response)

Reviewer #2: -

Reviewer #3: The objective of these analyses is to test new animal models for leishmania infection studies. The analysis followed as according to the study plan. A total of five species and 4 strains of one of the species, Leishmania martiniquensis were tested systemically in 3 rodent species, mice (BALB/c strain), steppe lemmings and Chinese hamsters. Results from the infections in the 3 infection animal models were presented logically with a comprehendable narrative. Negative infection rates were first presented for the BALB/c mice model, followed by asymptomatic infection demonstrated with the Chinese hamster model and symptomatic infection with the steppe lemming model. Comprehensive details in observations and analysis were included as part of the results which provided an in depth insight that is useful for future work using these animal models for Leishmania infection studies. Figures and tables summarise the data accurately and convey the message effectively for the purpose of these studies.

**Conclusions**

-Are the conclusions supported by the data presented?

-Are the limitations of analysis clearly described?

-Do the authors discuss how these data can be helpful to advance our understanding of the topic under study?

-Is public health relevance addressed?

Reviewer #1: (No Response)

Reviewer #2: -

Reviewer #3: The conclusions are supported by the data presented. Steppe lemmings and Chinese hamsters were demonstrated to be suitable animal models for Leishmania infection studies particularly for the Mudinia species while BALB/c mice were unsuitable as they were resistent to infection from these parasite species. Limitations for these analyses were mentioned in the discussion and suggested for further work that could be established based on results from these studies. 

**Editorial and Data Presentation Modifications?**

Reviewer #1: (No Response)

Reviewer #2: -

Reviewer #3: Line 216: (maximum 2,4x104) - check again if it is supposed to be 2,4 or 2.4, or 2-4?

Line 254: space between 88.9 and % symbol should be removed.

**Summary and General Comments**

Reviewer #1: The manuscript titled " Steppe lemmings and Chinese hamsters as new potential animal models for the study of the leishmania subgenus Mundinia (Kinetoplastida: Trypanosomatidae)" has been reviewed. The authors have effectively highlighted the potential of these species to serve as reservoir hosts for Mundinia. The manuscript holds value and contributes to the scientific community by introducing two new potential laboratory animal model species, Steppe Lemmings and Chinese Hamsters, for studying the leishmania subgenus Mundinia. However, the authors need to add the following queries for improvements.

-Line 73,75,79: replacing "Leishmania" with "L." and apply this change consistently throughout the manuscript.

-Line 119: Why did the authors choose to inject promastigotes into the left ear pinnae? Is this practice common in the literature or influenced by prevalence? Please discuss and consider including relevant references.

-Line 121 states that xenodiagnoses were performed at weeks 5, 10, 15, and 20 post-infection using Phlebotomus duboscqi. Why did the authors choose Ph. duboscqi for this study? Additionally, did the authors collect blood from the animals weekly for detecting Leishmania? Please provide a discussion on this matter.

-Line 171,181,206: Clarify the number of animals. Why are the conditions not consistent? Is there duplication in testing?

-Line 216: Replace 'maximum 2,4x10^4' with '2.4 x 10^4'."

Reviewer #2: The revisions of the manuscript are needed as mentioned above.

Reviewer #3: Overall, this manuscript is well written, easy to read and comprehend. The introduction provided sufficient insights into the topic about Leishmania as a pathogen and its significance to human health. The methods are written clearly with sufficient detail for replication of results. The study is novel and provides additional knowledge to the field, enabling scientists to further investigate infection related studies for Leishmania Mudinia subspecies, potentially enabling researchers to study new therapetics for this disease. I would like to question:

Regarding the rationale for the selection of animals for xenodiagnoses trials, it is only indicated for the BALB/c mice and steppe lemmings infection studies that 4 animals per group were selected for xenodiagnoses trial. However, it is not indicated in the section 3.2 Development of Mudinia in Chinese hamsters about the number of animals that had been selected for the xenodiagnoses trial. 

1) How many animals were chosen for each Leishmania species infection for the Chinese hamster xenodiagnoses trial? - to indicate this in the manuscript within the section.

2) Is there a reason why these 4 animals out of 10 animals in each group were selected? - to explain the reason behind how these animals were selected in the manuscipt.

PLOS authors have the option to publish the peer review history of their article (what does this mean?). If published, this will include your full peer review and any attached files.

Reviewer #1: No

Reviewer #2: No

Reviewer #3: No
---

## [Decision Letter · Decision Letter 1]

16 Apr 2024

Dear Dr. Sadlova,

We are pleased to inform you that your manuscript 'Steppe lemmings and Chinese hamsters as new potential animal models for the study of the Leishmania subgenus Mundinia (Kinetoplastida: Trypanosomatidae)' has been provisionally accepted for publication in PLOS Neglected Tropical Diseases.

Best regards,

Nadira D. Karunaweera

Academic Editor

Abhay Satoskar

Section Editor

Reviewer's Responses to Questions

**Key Review Criteria Required for Acceptance?**

**Methods**

-Are the objectives of the study clearly articulated with a clear testable hypothesis stated?

-Is the study design appropriate to address the stated objectives?

-Is the population clearly described and appropriate for the hypothesis being tested?

-Is the sample size sufficient to ensure adequate power to address the hypothesis being tested?

-Were correct statistical analysis used to support conclusions?

-Are there concerns about ethical or regulatory requirements being met?

Reviewer #1: (No Response)

Reviewer #2: (No Response)

Reviewer #3: All comments from all reviewers have been addressed accordingly.

**Results**

-Does the analysis presented match the analysis plan?

-Are the results clearly and completely presented?

-Are the figures (Tables, Images) of sufficient quality for clarity?

Reviewer #1: (No Response)

Reviewer #2: (No Response)

Reviewer #3: All comments from all reviewers have been addressed accordingly.

**Conclusions**

-Are the conclusions supported by the data presented?

-Are the limitations of analysis clearly described?

-Do the authors discuss how these data can be helpful to advance our understanding of the topic under study?

-Is public health relevance addressed?

Reviewer #1: (No Response)

Reviewer #2: (No Response)

Reviewer #3: All comments from all reviewers have been addressed accordingly.

**Editorial and Data Presentation Modifications?**

Reviewer #1: (No Response)

Reviewer #2: (No Response)

Reviewer #3: (No Response)

**Summary and General Comments**

Reviewer #1: (No Response)

Reviewer #2: The statement of the objective has been clarified. More methodological details and critical discussion with implications have been added, making the revised version much improved and more comprehensive. From my point of view, it is now ready to be accepted.

Congratulations!

Reviewer #3: All comments from all reviewers have been addressed accordingly.

PLOS authors have the option to publish the peer review history of their article (what does this mean?). If published, this will include your full peer review and any attached files.

Reviewer #1: No

Reviewer #2: **Yes: **Kanok Preativatanyou

Reviewer #3: No

---

## [Editor Report · Acceptance letter]

25 Apr 2024

Dear Dr. Sadlova,

We are delighted to inform you that your manuscript, "Steppe lemmings and Chinese hamsters as new potential animal models for the study of the *Leishmania* subgenus *Mundinia* (Kinetoplastida: Trypanosomatidae) ," has been formally accepted for publication in PLOS Neglected Tropical Diseases.

Best regards,

Shaden Kamhawi

co-Editor-in-Chief

Paul Brindley

co-Editor-in-Chief
